# Quantitative NMR Spectrometry of Phenylpropanoids, including Isoeugenol in Herbs, Spices, and Essential Oils

**DOI:** 10.3390/foods13050720

**Published:** 2024-02-27

**Authors:** Pascal Fabry, Sandra Weber, Jan Teipel, Elke Richling, Stephan G. Walch, Dirk W. Lachenmeier

**Affiliations:** 1Chemisches und Veterinäruntersuchungsamt (CVUA) Karlsruhe, Weissenburger Strasse 3, 76187 Karlsruhe, Germany; pas.fabry@t-online.de (P.F.); sandra.weber@cvuaka.bwl.de (S.W.); jan.teipel@cvuaka.bwl.de (J.T.); stephan.walch@cvuaka.bwl.de (S.G.W.); 2Department of Chemistry and Toxicology, RPTU Kaiserslautern-Landau, Erwin-Schrödinger-Strasse 52, 67663 Kaiserslautern, Germany; elke.richling@chem.rptu.de

**Keywords:** isoeugenol, NMR spectrometry, phenylpropanoids, herbs, spices, flavors, nutmeg, coffee, sweet flag

## Abstract

Isoeugenol (2-methoxy-4-(1-propenyl)phenol) has been recently classified as possibly carcinogenic to humans (Group 2B) by the International Agency for Research on Cancer (IARC). This study conducted an analysis of isoeugenol in common herbs and spices, including basil, cinnamon, ginger, and nutmeg, using ^1^H nuclear magnetic resonance (NMR) spectrometry. Additionally, over 1300 coffee samples were analysed by ^1^H-NMR for isoeugenol, but it was not detected in any of the analysed samples. Various essential oils, including nutmeg, basil, clove, sweet flag, and ylang-ylang oils, were examined for isoeugenol content. Out of the twelve nutmeg oils tested, four contained isoeugenol, with concentrations ranging from 3.68 ± 0.09 g/kg to 11.2 ± 0.10 g/kg. However, isoeugenol was not detected in the essential oils of calamus, basil, ylang-ylang, and clove using NMR spectrometry. These findings warrant critical evaluation of the previous literature, given reports of high isoeugenol levels in some of these matrices. A toxicological assessment has determined that there is no risk to human health by exposure to isoeugenol via nutmeg essential oils.

## 1. Introduction

Phenylpropanoids are an essential component of many plants. The compounds of this group serve as stress indicators, defensive substances against microorganisms, insects or mammals, and as attractants for pollinators [1]. Some of these compounds are suspected to be carcinogenic to humans, including methyleugenol [2], estragole, and safrole [3]. In 2010, the National Toxicological Program (NTP) performed a chronic toxicity and carcinogenicity study of isoeugenol (2-methoxy-4-prop-1-enylphenol) with groups of 50 male and female rats (F344/N) and mice (B6C3F1). The rodents were exposed to isoeugenol in corn oil by gavage. The NTP concluded that isoeugenol caused liver cancer in male mice. In addition, there was an increased incidence of histiocytic sarcomas in female mice and rare thyroid and mammary gland tumors in male rats [4]. Based on these data, the International Agency for Research on Cancer (IARC) recently published its evaluation of the substance isoeugenol. Isoeugenol was evaluated as a possibly carcinogenic substance and classified in Group 2B [2].

Isoeugenol is a viscous, yellowish liquid with a characteristic odor of clove and belongs to the phenylpropanoids. The substance was described to be occurring in over 500 plant species among others in nutmeg (*Myristica fragrans*), basil (*Ocimum basilicum*), sweet flag/calamus (*Acorus calamus*), bay leaf (*Laurus nobilis*) and coffee (*Coffea* spp.). High levels of isoeugenol were reported for nutmeg (40–320 mg/kg), the rhizome of calamus (228–12,510 mg/kg), and bay leaves (1000–6000 mg/kg) [5]. Isoeugenol and other phenylpropanoids are naturally synthesised in plants from the aromatic amino acid phenyl-alanine [6].

In previous studies, isoeugenol was quantified in the wine of the varieties Cabernet Sauvignon (6.49–7.74 µg/L) and Tempranillo (5.14–15.0 µg/L) [7]. The identification of isoeugenol in wine can be explained due to the extraction from oak barrels or added oak chips [5]. Moreover, two studies reported *trans*-isoeugenol in smoked sausages with levels of 8.67–29.03 mg/kg [8] and 6.4–75.6 mg/kg [9]. In comparison to the non-smoked sausages (4 mg/kg), significantly higher levels were detected. In addition to identification in food, isoeugenol was also detected in wood smoke. During the combustion of oak, eucalyptus, and pine wood, levels of 1.0, 0.5, and 17 mg/kg, respectively, have been determined [10]. The elevated isoeugenol levels in smoked sausages can thus be explained by the smoking process, whereby the pyrolysis of lignin leads to isoeugenol [5].

The substance is used as a flavoring agent in perfumes, cosmetics, and hygiene products [5]. Regulation (EC) No. 1223/2009 defines maximum concentrations for isoeugenol and the related compounds eugenol, methyleugenol, and safrole. Furthermore, isoeugenol is used in various foods as a flavoring agent, e.g., in baked goods. In addition, isoeugenol finds application as an anesthetic for fish [11] as well as a starting compound for vanillin synthesis [12].

Due to the diverse occurrence of isoeugenol, different quantification methods have been suggested for its analysis, including high-performance liquid chromatography (HPLC) [13,14,15] or gas chromatography–tandem mass spectrometry (GC-MS/MS) [11]. These chromatographic analyses, characterised by their high sensitivity, contrast with the advantages offered by ^1^H nuclear magnetic resonance spectrometry (NMR), such as reproducibility and reduced analysis time, and broader applicative flexibility. Despite this, no studies have been found that use ^1^H nuclear magnetic resonance spectrometry (NMR) to distinguish isoeugenol from structurally related derivatives. Also, there are no published quantitative ^1^H-NMR methods for detecting isoeugenol. The main objective of this research is to evaluate the feasibility of identifying isoeugenol in commonly used herbs, spices, and flavorings via ^1^H-NMR. Additionally, the study will explore the differentiation of isoeugenol from its structurally similar derivatives, including eugenol, methyleugenol, estragole, safrole, α- and β-asarone. Subsequently, a toxicological assessment of the detected isoeugenol concentrations will be conducted.

## 2. Materials and Methods

### 2.1. Samples

The examined herbs and spices were commercially purchased in shops in the vicinity of Karlsruhe, Germany, in the period from April 2023 to July 2023. The essential oils were also commercially purchased during the same period via online markets from various countries, including Germany, the United Kingdom, and India.

The samples included nutmeg (*Myristica fragrans*), with 10 samples of its seeds and 1 sample of its mace, basil (*Ocimum basilicum*) with 4 samples of leaves, tarragon (*Artemisia dracunculus*) with 1 sample of leaves, bay leaf (*Laurus nobilis*) with 1 sample of leaves, dill (*Anethum graveolens*) with 1 sample of leaves, annual mugwort (*Artemisia annua*) with one sample of leaves, ginger (*Zingiber officinale*) with 2 samples of rhizomes, cinnamon (*C. verum/aromaticum*) with 3 samples of barks, clove (*Syzygium aromaticum*) with 1 sample of fruit, allspice (*Pimenta dioica*) with 1 sample of fruit, and calamus (*Acorus calamus*) with 5 samples of rhizomes. Green tea leaves (*Camellia sinensis*) and fennel tea seeds (*Foeniculum vulgare*) were each investigated with two samples, while an extensive dataset of roasted seed samples (*Coffea arabica* and *C. canephora*) was studied using 1326 ^1^H-NMR spectra.

Additionally, a comprehensive examination of the essential oils obtained from different plants was carried out. The samples included nutmeg (*Myristica fragrans*) with 12 samples of essential oil extracted from seeds (9 commercial samples and three self-distilled samples, see Section 2.2.), basil (*Ocimum basilicum*) with 4 samples from leaves, calamus (*Acorus calamus*) with 5 samples from rhizomes, clove (*Syzygium aromaticum*) with 4 samples from fruits and ylang-ylang (*Cananga odorata*) with 2 samples from flowers. In addition to that, perilla (*Perilla frutescens*) with one sample of cooking oil extracted from flowers, as well as one sample of liquid smoke flavor, was analysed.

### 2.2. Sample Preparation

The dried herbs and spices were ground to a fine powder using a cutting mill (A11 basic, IKA, Staufen, Germany). Already powdered samples were used without further grinding. Fresh sample material was ground in a blender (La Moulinette XXL, Moulinex, Frankfurt am Main, Germany) to a homogeneous material. Sample material not required immediately was stored under a nitrogen atmosphere at 3 °C to protect it against oxidation. Following that, the sample preparation according to Monakhova et al. was used [16]. In short, 200 mg of sample material was mixed with 1.5 mL (^2^H)chloroform, stabilised according to Teipel et al. [17], and shaken mechanically (Edmund Bühler, Bodelshausen, Germany) at 350 rpm for 20 min. The extraction agent and the extraction time were varied and suitable conditions for the respective sample were determined. To optimise the extraction time, all sample matrices were processed in duplicate, testing extraction times of 5, 45, 60, 90, and, if necessary, 120 min. The extraction was carried out using a mechanical shaker.

(^2^H)chloroform (Carl Roth, Karlsruhe, Germany) and (^2^H_4_)methanol (Merck, Darmstadt, Germany) were used as possible extraction agents. The suspension was passed through a membrane filter (0.2 µm). From the obtained solution, 600 µL was analysed by ^1^H NMR spectrometry.

In the case of the analysed coffee samples, about 10 g of roasted coffee beans were ground in a coffee grinder (Hemro, Zurich, Switzerland) to a standardised particle size. The subsequent processing, including extraction of 200 mg ground coffee with 1.5 mL (^2^H)chloroform, was carried out analogous to the study by Monakhova et al. [16].

The nutmeg samples (*Myristica fragrans*) were extracted with (^2^H_4_)methanol for a duration of 60 min. For sweet flag (*Acorus calamus*), the extraction was carried out using (^2^H)chloroform for 45 min. Basil (*Ocimum basilicum*) was subjected to (^2^H)chloroform extraction for 60 min.

Additionally, bay leaf (*Laurus nobilis*), tarragon (*Artemisia dracunculus*), and clove (*Syzygium aromaticum*) were investigated using (^2^H)chloroform extraction. The extraction durations for these samples were 60 min for bay leaf, 45 min for tarragon, and 45 min for clove. The extraction agent of allspice (*Pimenta dioica*) was (^2^H_4_)methanol and the time of extraction was 90 min.

For essential oil analysis, 50 mg of the sample was dissolved in 1 mL of (^2^H)chloroform, and 600 µL was transferred into an NMR tube and analysed by ^1^H NMR spectrometry. The essential oils were stored at room temperature in the absence of light.

As an additional method for the sample preparation of nutmeg, its powder was subjected to steam distillation to obtain nutmeg essential oil. This process was based on the specifications in DIN EN ISO 6571 [18], which, however, is designed only for the volumetric determination of the essential oil and not for further analysis using NMR. Because of this, a modified methodology for subsequent NMR measurements was investigated. The sample weight was doubled from 7.5 g to 15 g, the volume of water was increased from 200 mL to 400 mL, the duration of distillation was extended from 3 h to 4 h, and (^2^H_10_)p-xylene was used as the solvent for the essential oil instead of non-deuterated p-xylene. The distillate obtained was transferred to a brown HPLC vial and stored in a nitrogen atmosphere analogous to commercially obtained essential oils. The analysis was identical to that of the commercially purchased oils. Of the 10 nutmeg samples, only three could be used for steam distillation because of an insufficient amount of material.

### 2.3. Chemicals

The analytes to be investigated were purchased commercially. All standards used are of p.a. (*pro analysi*) quality. A mixture of *cis*- and *trans*-isoeugenol was used as the standard for isoeugenol. However, separation of *cis*- and *trans*-isoeugenol could not be achieved by NMR, so the contents were determined as the sum of *cis*- and *trans*-isoeugenol. The standards (*c* = 10 mg/mL) were characterised by ^1^H-NMR spectrometry. All standards were stored under an argon atmosphere in a refrigerator at 3 °C to protect them from oxidation.

The standards used were isoeugenol (CAS RN. 97-54-1; Sigma-Aldrich, St. Louis, MO, USA); eugenol (CAS RN. 97-53-0; Sigma-Aldrich, St. Louis, MO, USA); methyleugenol (CAS RN. 93-15-2; Sigma-Aldrich, St. Louis, MO, USA); estragole (CAS RN. 140-67-0; Sigma-Aldrich, St. Louis, MO, USA); safrole (CAS RN. 94-59-7; Carl Roth, Karlsruhe, Germany); α-asarone (CAS RN. 2883-98-9; Carl Roth, Karlsruhe, Germany); and β-asarone (CAS RN. 5273-86-9; Carl Roth, Karlsruhe, Germany).

### 2.4. ^1^H NMR Measurements

All measurements were performed on a 400 MHz NMR spectrometer (NMR Ascend 400, Bruker BioSpin, Ettlingen, Germany) with a Bruker broadband inverse probe and a Bruker Sample Xpress sample changer. The parameters used for the respective solvents are listed in Table 1. For the solvent (^2^H_4_)methanol an additional water suppression was added. The measuring times in (^2^H_4_)methanol and (^2^H)chloroform were approximately 30 and 40 min, respectively. For the ^1^H qNMR analyses, DeuQuant NMR tubes (diameter 5 mm) from Deutero (Kastellaun, Germany) were used. The inner diameter of the NMR tubes had to be identical for each measurement; otherwise, fluctuations in the determined contents were to be expected [19]. Qualitative measurements were performed using disposable polyethylene (PE) caps, whereas all quantitative measurements were performed with polytetrafluoroethylene (PTFE) caps, effectively minimising solvent evaporation.

For the identification of the analytes, one-dimensional (1D) ^1^H NMR spectra were recorded with a 90° pulse. In addition, the spectra of isoeugenol in the solvents (^2^H_4_)methanol and (^2^H)chloroform were measured as J-resolved experiments (JRES). The two-dimensional (2D) JRES allows the evaluation of complex spectra that are difficult or impossible to separate under 1D conditions due to overlapped signals or cannot be assigned unambiguously under 1D conditions. Along the x-axis, the spectrum is displayed in ppm analogous to the 1D measurement, and, along the y-axis, the multiplicities of the signals are given in Hz [20].

### 2.5. Quantification

Due to the proportionality between signal intensity and the number of excited protons, quantification by NMR is possible. One method for this is the Pulse Length Based Concentration (PULCON) method. This method uses an external reference sample with a known concentration, which is measured on the same NMR spectrometer with identical parameters as the sample of unknown concentration. After baseline and phase correction, a factor is determined from the signals obtained from the reference. This factor can be used for the calculation of the analytes of the sample.

For the preparation of the external reference, also called QuantRef (Quantification Reference Solution), the molar masses of the substances used, as well as their weighed portions and purities, are required. Furthermore, the exact position of the signals and the number of protons behind them are required [21]. For the quantifications in (^2^H)chloroform, the used QuantRef consisted of 1,2,4,5-tetrachloronitrobenzene (TCNB) (7.83–7.63 ppm), p-xylene (6.96–7.16 ppm and 2.18–2.43 ppm) and cyclohexane (1.30–1.45 ppm). For the quantifications in (^2^H_4_)methanol, the used QuantRef consisted of phenoxyethanol (6.85–6.99 ppm; 3.94–4.11 ppm and 3.71–3.93 ppm) and ethylbenzene (7.08–7.19 ppm; 2.53–2.71 ppm and 1.10–1.30 ppm). For quality control monitoring of the analyses, the recovery rate of a control solution was measured at the end of each series. For measurements in (^2^H)chloroform, TCNB was used as the control substance. For (^2^H_4_)methanol measurements, nicotinamide was used.

### 2.6. NMR Spectral Processing

A MATLAB v. 2019b (The MathWorks, Natick, MA, USA) script was created for each solvent to achieve a fast and precise quantitative evaluation of the ^1^H NMR spectra. For the quantification of isoeugenol, the signals in Table 2 were used. Depending on the matrix, overlaps of the signals to be quantified occurred. It was necessary to confirm the quantification with another signal range.

### 2.7. Limit of Detection and Quantification

The limit of detection (LOD) and limit of quantification (LOQ) were determined by linear regression based on five concentration points. The lowest, medium, and highest concentrations were measured in triplicate. LOD and LOQ were calculated according to DIN 32645:2008 [22].

## 3. Results and Discussion

### 3.1. Method Development and Optimisation

#### 3.1.1. Solvent Selection

No reports of previous analyses of isoeugenol by NMR spectrometry have been found in the literature. Therefore, different solvents were initially investigated for their suitability for the separation of isoeugenol from structurally related phenylpropanoids. In addition, it was examined whether the respective solvents are suitable for liquid extraction with subsequent filtration. The solvents used were (^2^H_4_)methanol, (^2^H)chloroform, and (^2^H_6_)dimethyl sulfoxide (DMSO). Since a wide variety of solvents were used in published studies for the extraction of isoeugenol, a large proportion of the eluotropic series could be covered. For the different sample matrices with their different contents of polar and apolar ingredients, it was thus possible to select the solvent that extracted the least amount of interfering substances.

First, spectra of isoeugenol and its derivatives were recorded in methanol, chloroform, and dimethyl sulfoxide (*c* = 10 mg/mL). In spectra with (^2^H_4_)methanol as a solvent, the signals of the methoxy protons of isoeugenol, eugenol, methyleugenol, and estragole show clearly separate chemical shifts (Figure 1).

Isoeugenol could also be distinguished from its derivates in spectra of solutions in (^2^H)chloroform. The signals from eugenol and methyleugenol slightly overlap, making quantification of all the phenylpropanoids difficult (Figure 2).

(^2^H_6_)DMSO as solvent enabled the differentiation between isoeugenol and its derivatives, but it proved ineffective in separating the further phenylpropanoids. Furthermore, (^2^H_6_)DMSO caused blockages in membrane filters in several samples, rendering it unsuitable. Consequently, (^2^H_6_)DMSO was excluded as a solvent choice, leading to the selection of (^2^H_4_)methanol and (^2^H)chloroform for further investigations.

#### 3.1.2. Identification of Isoeugenol

Figure 3 shows the spectrum of isoeugenol in (^2^H_4_)methanol. Deshielded by the aromatic ring current effect, protons B, C, and D give signals in the downfield region of the spectrum. These three protons can be differentiated due to their coupling constants. Doublet B has a coupling constant of 1.8 Hz, which indicates a meta-coupling with an aromatic proton. Doublet of doublet C also has a coupling constant of 1.8 Hz and additionally one of 8.2 Hz, which indicates ortho-coupling with an aromatic proton [23]. Doublet D also has a coupling constant of 8.2 Hz. These numbers indicate that proton B is the lone proton (H-3) between the methoxy group and the alkyl side chain and forms a ^4^J coupling to proton C. Proton C (H-5) can be identified because of its coupling constants to both ortho- and meta-aromatic protons. Proton D (H-6) results from the identical coupling constant with proton C. The JRES spectrum of isoeugenol in (^2^H_4_)methanol (Figure 4) verifies the assumption that proton B is the lone proton, since the proton at 6.91 ppm has no direct neighbors.

For the two subsequent signals at 6.29 and 6.07 ppm a coupling constant of 15.6 Hz was determined, indicating that these are the protons of the double bond. Since proton F splits into a doublet of quartets, this signal originates from the proton next to the methyl group. Signal E, which also splits into a doublet of quartets, is therefore the proton of the double bond directly next to the aromatic ring. The following singlet G at 3.84 ppm is the signal of the methoxy group. Signal H at 1.86 ppm is composed of three protons and splits into a doublet of doublets with the same coupling constants as signals E and F. Therefore, signal H is the methyl group at the end of the alkyl side chain.

The assignment of the isoeugenol spectrum in (^2^H)chloroform (Figure 5) turned out to be more complicated than its spectrum recorded in (^2^H_4_)methanol, since no clean coupling constant could be obtained from the signals of the aromatic protons because the coupling constants and shift differences of the aromatic protons are of similar size (higher order spectrum). In addition, the given spectrum contains one coherent signal for two of the aromatic protons, which means that further information for the exact assignment of the signals was lost. From the 1D spectrum of isoeugenol in (^2^H)chloroform, it is not possible to make a justified assignment of the aromatic signals to the respective protons. Indications of a possible assignment of the signals are provided by the recorded JRES spectrum (Figure 6). Signal B at 6.86 ppm indicates that there is another proton located in the immediate vicinity of this proton. Thus, this signal could be assigned to the aromatic proton next to the hydroxyl group. Consequently, the two remaining aromatic protons (signal C) could interact with each other via a ^4^J coupling. A reverse assignment of the neighboring protons with resulting ^5^J coupling via the para-positioned aromatic atoms is considered less likely. Unlike the isoeugenol spectrum in (^2^H_4_)methanol, the spectrum of (^2^H)chloroform provided the signal of the hydroxy group proton at 5.53 ppm. All further assignments were analogous to those in (^2^H_4_)methanol.

#### 3.1.3. Optimisation of Extraction

To evaluate the quality of extraction, derivatives of isoeugenol were examined instead of isoeugenol itself, as isoeugenol was not detectable in any spice or herbal sample. The initially considered extraction method of ultrasound-assisted liquid extraction was abandoned because of the formation of an unidentifiable precipitate in the methanol extraction of nutmeg that affected the signal intensities. An extraction time for the respective sample matrices was set once the analyte contents increased by less than 10% with further extension.

Nutmeg, with 30–40% fat and about 10% essential oil content, responds differently to extraction methods. Extraction with (^2^H_4_)methanol better separates analytes like safrole and methyleugenol from its complex matrix, which includes terpene derivatives (α-pinene, camphene, p-cymene) and phenylpropanoids (myristicin, elmicin) [24]. However, (^2^H_4_)methanol extraction is complicated by nutmeg’s high sugar content [25], leading to significant matrix effects. On the other hand, the non-polar (^2^H)chloroform increases the extraction of lipophilic substances but is less effective for polar compounds. Given nutmeg’s complexity, with both polar and non-polar compounds, steam distillation was used for identifying isoeugenol via ^1^H NMR spectrometry.

Phenylpropanoids in basil, tarragon, bay leaf, and calamus were extracted more effectively with chloroform than with methanol. Contrary to initial assumptions based on calamus’s composition (10% glycolipid, 2.5% sterol, 0.33% free fatty acids, and 2.25% total sugars, including 0.85% sucrose and 1.08% fructose) [26], experimental studies revealed that chloroform, not methanol, was more efficient for extraction. This finding was in line with the compositions of the other herbs [27,28,29,30], where less polar chloroform outperformed methanol in extraction, as confirmed by subsequent experiments.

For clove extraction, (^2^H)chloroform was chosen as the solvent for a 45 min extraction period, yielding higher eugenol content. For allspice, (^2^H4)methanol proved more efficient, with a 90 min extraction time significantly increasing the yield of eugenol and methyleugenol.

#### 3.1.4. Steam Distillation of Nutmeg

Regarding the complex matrix of nutmeg, steam distillation was investigated as an alternative method to solvent extraction. It is an elegant method of separating the essential oils from the rest of the plant material [31]. Four distillation variants for nutmeg were tested, one of which used the parameters of DIN EN ISO 6571:2018 [18] and three used a modified version. By the use of the DIN EN ISO 6571:2018 [18] method, a significant improvement in signal strength and sharpness was observed in comparison to solvent extraction.

Using the modified parameters, three essential oils were prepared. In two of these samples, isoeugenol was detected by ^1^H NMR spectrometry. The spectra of one nutmeg sample containing isoeugenol are shown in Figure 7 using both extraction methodologies, steam distillation as well as solvent extraction. The spectrum obtained via steam distillation shows sharper signals and clearly recognizable signals of isoeugenol in contrast to the spectrum of solvent extraction. This observation can be explained by the higher concentration of isoeugenol in nutmeg essential oil. Since nutmeg consists of about 10% essential oil [24], steam distillation results in an enrichment of the essential oil constituents by a factor of 10. NMR spectrometry, due to its quantum mechanical properties, has a lower sensitivity compared to other methods, so the low content of isoeugenol could not be detected by NMR in nutmeg samples prepared with solvent extraction. By increasing the concentration by steam distillation, this problem can be circumvented. Thus, steam distillation provides a suitable method for the identification of isoeugenol in nutmeg.

#### 3.1.5. Method Validation Results

A list of the LODs and LOQs of the selected matrices is presented in Table 3.

Coffee spiked with isoeugenol and extracted with (^2^H)chloroform [32] showed an overdetermination of around 110% at concentrations under 44.3 to 222 mg/kg. However, the expected levels were reached at higher concentrations (355 mg/kg to 443 mg/kg) with approximately 100.3 ± 1.3% recovery. For the spiking of calamusprior to extraction with (^2^H)chloroform, the recovery at concentrations under 448 mg/kg to 2243 mg/kg were close to the target values (98.9 ± 5.1% and 99.4 ± 1.9%); however, lower recoveries were determined for concentrations at 3585 mg/kg (93.4%) and 4478 mg/kg (93.8 ± 1.7%). In nutmeg essential oil, a constant recovery rate of about 95% (94.7 ± 1.09% to 95.9 ± 3.9%) was measured in (^2^H)chloroform at concentrations between 822 mg/kg and 8220 mg/kg. Overall, the recoveries of these three experiments were within an acceptable range.

Consequently, the recovery rates of the spiking experiments were in the expected range. The method validation proved the applicability of the methods for the purpose of identifying and quantifying isoeugenol using NMR spectrometry.

### 3.2. Results of Solid Samples

Ten nutmegs were analysed, seven whole nutmegs, which were ground with a knife mill, and three commercially purchased samples of nutmeg powder. Additionally, one ground mace was also examined. Safrole was detected in 10 of 11 samples and methyleugenol could be identified in 4 of 11 nutmeg samples. These results agree with the study by Dupuy et al. [33], who analysed essential oils of nutmeg. In this study, besides myristicin, methyleugenol, and safrole were the main components of the phenylpropanoids.

In addition, a total of eight herbs were examined, including three fresh basils, one dried basil, as well as tarragon, bay leaf, dill, and annual mugwort, each in dried form. In all fresh basils, methyleugenol was identified, as well as eugenol in one sample. Estragole was also detected in the dried basil. A study by Miele et al. [34] on the analysis of phenylpropanoids in basil also concluded that methyleugenol and eugenol are the main aromatic constituents. In another study, estragole was found to be the major constituent of the essential oil of basil [35].

Tarragon was found to contain estragole and methyleugenol, which are also listed in the literature as the main components of phenylpropanoids [36]. Analogous to Nenadis et al. [37], the analytes eugenol and methyleugenol were detected in the bay leaf.

None of the analytes were identified in both dill and annual mugwort. The essential oil of dill is predominantly composed of α-phellandrene, myristicin, and dill ether [38]. The major components of the essential oil of annual mugwort are camphor, 1,8-cineole, and germacrene D [39].

None of the seven analytes could be identified in three different barks of cinnamon. The essential oil of cinnamon consists of more than 80% cinnamaldehyde [40]. According to the same study, eugenol should also be present in cinnamon essential oil; however, its content was apparently too low to obtain detectable signals.

Likewise, none of the analytes could be detected in fresh as well as ground ginger. The essential oil of ginger consists predominantly of α-zingiberene, β-bisabolol, and geranial [41].

In cloves, the high contents of eugenol described in the literature could be verified. In allspice, the high contents of eugenol and methyleugenol could also be verified [42,43].

Furthermore, five different teas from the rhizome of calamus were analysed by NMR spectrometry. In all five samples, only β-asarone was identified. This result verifies previous studies on calamus, which report contents of β-asarone in the essential oil of more than 80% [44].

Phenylpropanoids could not be detected in either of the two green teas examined. The solvent extraction yielded low levels of estragole in both fennel teas analysed. A study by Rather et al. [45] identified estragole in fennel, which, along with *trans*-anethole, is the main component of aromatic substances.

### 3.3. Results in Essential Oils

Since no isoeugenol could be identified in the previous analyses of solid samples, the focus was shifted to the study of essential oils. Because phenylpropanoids occur in high quantities in essential oils, a concentration of the respective analytes can thus be achieved [46]. A total of nine essential oils of nutmeg were commercially available and three essential oils were prepared from plant material by steam distillation (see Section 2.2). Nine of the twelve nutmeg oils contained safrole and methyleugenol. These analytes accounted for the largest proportion of the essential oil if they were present. One exception occurred, in which estragole made up the highest proportion. Eugenol occurred in 9 out of 12 samples. Isoeugenol was identified in two commercial and two distilled essential oils.

Of the five calamus oils measured, four were composed of β- and α-asarone. In one calamus oil, none of the phenylpropanoid analytes were identified. Previous studies described a proportion of more than 80% β-asarone and almost 10% α-asarone in the essential oil of calamus rhizome [44], which corresponds approximately to the concentration ratios in the NMR measurements. Only one exceptional sample contained almost equal ratios of β- and α-asarone.

Estragole was detected in all four basil essential oils analysed, which, according to previous studies, constitutes the main component of the essential oil of basil [35]. Additionally, three samples contained eugenol, and two samples contained methyleugenol.

Furthermore, four different clove oils were analysed. High concentrations of eugenol were detected in the examined samples, which corresponds to the findings of previous studies [42].

In addition, two ylang-ylang oils and one perilla cooking oil were analysed. Eugenol was identified in one ylang-ylang sample, which was also reported by other authors [47]. According to this study, other components are germacrene D and α-farnesene. None of the phenylpropanoids could be detected in the other ylang-ylang oil and the perilla cooking oil. The main components in perilla oil, according to the literature, are carvone, perilla aldehyde, and caryophyllenes [48]. No analytes were identified in the liquid smoke flavor.

### 3.4. Quantitative NMR Measurement of the Essential Oil of Nutmeg

The contents of isoeugenol in the essential oils of nutmeg are listed in Table 4. In many cases, the correct signals of isoeugenol could only be verified by prior spiking. Due to the possibility of integrating several signals by NMR, the most suitable signals for quantification could be found. Figure 8 shows the singlet at 3.89 ppm of sample 1.

In a study by Dupuy et al., nutmeg essential oils from Indonesia were analysed by GC-MS [33]. The 14 nutmeg samples were distilled for 9 h and *trans*-isoeugenol contents ranging from 0.04 to 1.30% were determined. In addition, the authors were also able to identify differences in phenylpropanoid content due to the maturity of the nutmegs. Unripe nutmegs contained mainly terpenes, while ripe nutmegs contained high concentrations of phenylpropanoids. In a series of experiments with ripe nutmegs, isoeugenol contents of 0.36–2.36% were determined.

In another study, nutmeg essential oils from Indonesia were also analysed by GC-MS [49]. Steam distillation was used to produce essential oil from 20 nutmegs of varying maturity. In addition, care was taken to separate oil from the seed and the nutmeg flower to obtain a comparison of these two plant parts. The highest contents were analysed in the mace (1.38 ± 0.12%). Analogous to the study of Dupuy et al., it was found that the contents of isoeugenol of young mace (0.44 ± 0.18%), to medium ripe nutmegs (0.98 ± 0.05%), to old nutmegs (1.21 ± 0.00%) increased noticeably. Similarly, it was also observed that the contents of terpene compounds decreased with increasing age and the contents of phenylpropanoids increased.

In this paper, contents between 0.37 ± 0.01% and 1.12 ± 0.01% were calculated, placing the results in the same order of magnitude as the two previous studies. Sample 2 with the lowest content of 0.37%, originated from Sri Lanka. Sample 1 with the highest content of 1.12% comes from India. Sample 4 (0.84%), which was steam distilled, comes from Indonesia. Sample 3 (0.84%), which was also self-distilled, has an unknown country of origin.

### 3.5. Isoeugenol in Coffee

Numerous studies have already reported the identification and quantification of isoeugenol in various types of coffee by GC-MS. Wu and Cadwallader investigated isoeugenol in Chicory coffee and calculated levels of 45.1 µg/L *trans*-isoeugenol and 15.4 µg/L *cis*-isoeugenol [50]. In *C. arabica* samples from Brazil, 2.141 µg/L isoeugenol was determined, while samples from Colombia contained 1.691 µg/L isoeugenol [51]. Other publications detected 0.81 µg/kg isoeugenol in Arabica coffee, and 0.12 mg/kg isoeugenol in roasted Colombian coffee [52,53].

Over 1300 NMR spectra of coffee samples measured between 2018 to 2023 were re-analysed for isoeugenol using MATLAB. For this purpose, the singlet of isoeugenol at 3.89 ppm was used as a marker, as this signal has the highest intensity and therefore the highest chance of identification. In addition, the median of the background noise of all measured coffee samples was calculated. Subsequently, a search was made for signals in the range of 3.89 ppm that exceeded the median by a factor of three. This factor was chosen based on the specifications of EuroLab Technical Report No. 01/2014, according to which a signal-to-noise ratio of three represents the mathematical detection limit [54]. Out of more than 1300 spectra, this factor was exceeded for 22 samples. These samples were manually analysed for isoeugenol. In these samples, isoeugenol could not be identified (LOD 20 mg/kg).

### 3.6. Discrepancy with Previous Literature Information

Eleven nutmegs were examined by solvent extraction and subsequent ^1^H qNMR analysis. In accordance with DIN 32645:2008 [22], a LOD of 191 mg/kg and a LOQ of 468 mg/kg was determined. The literature values of isoeugenol in nutmeg range from 40 to 320 mg/kg [5]. An identification of isoeugenol in nutmeg could not be provided. However, isoeugenol could be detected in two commercial and two self-made essential nutmeg oils. The findings of isoeugenol in essential oils of nutmeg can be explained by a concentration of the volatile constituents by a factor of about ten compared to the original nutmeg [24]. This shows that the LOD of the solvent extraction used was too high and thus isoeugenol was not identified by ^1^H NMR spectrometry. Consequently, the detection of isoeugenol in nutmeg using NMR requires prior concentration, for example, by steam distillation.

Contents between 3680 and 11,200 mg/kg were determined in the essential oils of nutmeg. According to IARC, contents of 1000–3000 mg/kg can be expected [5]. All contents of the NMR measurement are above these values. However, there are two studies from Dupuy et al. and Saputro et al. that report contents between 400 and 13,000 mg/kg and 4400 and 13,800 mg/kg, respectively, which are consistent with the ranges determined in this study. The figures of IARC refer to Duke [55], who provides no information on the analytical method used. The studies by Dupuy et al. and Saputro et al., on the other hand, are transparent in their methodology [33,49].

In addition to nutmeg, five teas made from calamus rhizome, as well as five calamus essential oils, were analysed. For isoeugenol, a LOD of 160 mg/kg and a LOQ of 386 mg/kg were determined for the solvent extraction with subsequent ^1^H NMR analysis. In the literature, high levels of isoeugenol are described in the rhizome of calamus, ranging from 228 to 12,510 mg/kg [5]. Consequently, isoeugenol in calamus should have been identified if the ranges from the literature were correct.

The volume of essential oil in calamus rhizome is 0.9% in relation to the weighed amount of calamus [44]. Thus, the extraction of essential oil from the rhizome increases the concentrations of the volatile compounds by a factor of about 100. Nevertheless, neither in five teas nor in five essential oils isoeugenol could be detected. Since isoeugenol could not be detected in ten different calamus samples, the available literature must be critically examined. The literature on isoeugenol in calamus refers exclusively to the publication by Duke, which, however, does not provide any information on the method used to determine the isoeugenol content [55]. Furthermore, no other studies on the identification of isoeugenol in calamus are known.

In addition, four basil samples (three fresh, one dried) and four basil essential oils were examined. Due to the same extraction agent and comparable spectra in the region of the quantifiable isoeugenol peaks, the figures determined for calamus can be assumed as an estimate of the LOD and LOQ (160 mg/kg and 386 mg/kg). The contents of isoeugenol in basil are 8–95 mg/kg and therefore too low to be analysed by solvent extraction with subsequent ^1^H NMR spectrometry. Consequently, isoeugenol was not identified in any of the four basil species. The essential oil content in basil, depending on the production process, ranges from 0.08 to 0.68% [56], which increases the concentrations of the constituents by a factor of over 100. Accordingly, the concentrations of isoeugenol in basil essential oil should be in a detectable range for NMR. However, no isoeugenol concentrations were found in any of the four basil essential oils. Analogous to calamus, the noted content levels of isoeugenol in basil originate from the publication by Duke, which again gives no indication of the analytical method used [55]. An alternative study that quantifies isoeugenol in basil was not found. However, a qualitative study detected isoeugenol in basil [57].

With regard to 1.48–1.68 mg/kg in the rhizome of ginger, 1–45 mg/kg in the leaves of annual mugwort, as well as 2–8 mg/kg in the bark of Ceylon cinnamon, isoeugenol is, in terms of the determined LODs (Table 3), not detectable in these matrices by ^1^H NMR spectrometry [5].

A previous study analysed the volatile compounds in bay leaf by GC-MS and determined contents of 1000–6000 mg/kg isoeugenol [58]. In another study, the essential oil of bay leaf was also analysed by GC-MS, and 3000 mg/kg *trans*-isoeugenol was detected [59]. Considering the proportion of essential oil in bay leaves (0.4–1.1%) [60], the contents of isoeugenol in the original bay leaves range between 12 and 33 mg/kg. As an estimate of the LOD and LOQ, the figures of calamus can be used (160 mg/kg and 386 mg/kg) due to the same extraction agent and similar spectra. Thus, these values are below the LOD of isoeugenol and cannot be detected by ^1^H NMR spectrometry. Confirming these estimations, isoeugenol was not found in the sample analysed.

According to the literature, the essential oil of perilla leaves contains high concentrations of isoeugenol of 2500 mg/kg [5]. However, no analytes were identified in the analysed cooking oil of perilla. This result can be explained by the fact that the used oil originates from the seed of *Perilla frutescens* and not from the leaves. No oils could be obtained from the leaves of the perilla plant at the available suppliers.

The manufacturer of one ylang-ylang essential oil reports eugenol contents of 0.45% and isoeugenol contents of 0.54% [61]. However, neither eugenol nor isoeugenol could be identified by ^1^H NMR spectrometry. The safety data sheet of the manufacturer does not contain any information on the method used to quantify the two analytes. Considering that using ^1^H NMR spectrometry, isoeugenol contents of approximately 0.37% were detected in nutmeg essential oil, the statement of the manufacturer must be critically questioned.

According to the literature, in addition to the main component eugenol (85–95%), clove oil also contains 0.5% isoeugenol [11]. However, no isoeugenol was identified in the four clove essential oils examined by ^1^H NMR spectrometry, which means that the claim about isoeugenol content in clove must also be questioned.

### 3.7. Evaluating the Risk of Isoeugenol Exposure from Nutmeg and Coffee Consumption

The Committee for Veterinary Medicinal Products (CVMP) announced an Acceptable Daily Intake (ADI) of 0.075 mg/kg body weight (bw) [62] based on the NTP study [4].

According to the literature, the daily consumption of nutmeg essential oil is 0.2245 mg/kg bw (based on 12% essential oil content) [63]. Assuming the highest isoeugenol content determined (1.12%), the potentially ingested level of isoeugenol is about 30 times lower than the ADI. Therefore, a 70 kg person would have to consume approximately 3.9 g/day of nutmeg to reach the ADI. In addition, it must be mentioned that the stated proportions of 12% essential oil in nutmeg are to be estimated quite high. According to ISO 6577:2002, the proportion of essential oil in nutmeg is only 6.5%, and other publications report about 10% [24,64].

A previous study investigated the consumption of herbs and spices in different European countries [65]. According to this study, the highest consumption was in Hungary with 1.54 g/day. Even if this consumption consists exclusively of nutmeg that has an essential oil content of 12% and the highest isoeugenol content determined in this paper (1.12%) the daily ingestion in this worst-case assumption would be 2.5 times lower than the ADI (for a 70 kg person). In view of the available data, it can be concluded that the isoeugenol content in nutmeg does not pose a risk to human health. This is similar to previous research on methyleugenol [66].

In the case of coffee, a LOD of 20 mg/kg was determined for isoeugenol. Assuming in a worst-case scenario that levels of isoeugenol at this detection limit are present and a 70 kg person consuming one cup of coffee per day (about 0.25 L), this would result in a consumption below the ADI by a factor of 0.9. A risk for consumers can therefore already be largely excluded with the relatively insensitive ^1^H NMR method based on a very large sample collective. The actual contents measured in coffee are typically well below 1 mg/kg, so the risk is becoming even smaller [50,51,52,53].

In general, this research corroborates another risk assessment of isoeugenol based on benchmark dose–response modelling, which provides evidence that the estimated daily exposure levels of isoeugenol are associated with only a low risk for consumers [67].

## 4. Conclusions

A variety of different herbs and spices were investigated using ^1^H NMR spectrometry. In addition, the ^1^H NMR spectra of over 1300 coffee samples were examined. In these samples, isoeugenol was not identified, so the focus was shifted to the analysis of essential oils. Since the phenylpropanoids, to which isoeugenol belongs, are present in high amounts in essential oils, the low sensitivity of NMR could be compensated. In four out of twelve nutmeg essential oils, isoeugenol was identified and quantified. Contents between 3.68 ± 0.09 g/kg and 11.2 ± 0.10 g/kg were determined. In the essential oils of calamus, basil, ylang-ylang, as well as clove, isoeugenol could not be identified. Consequently, the literature should be critically examined.

It was shown that ^1^H NMR spectrometry is a rapid and reproducible method for the analysis of isoeugenol and its structurally related phenylpropanoids. NMR’s foremost advantage, therefore, lies in its simplicity, efficiency, and effectiveness in detecting toxicologically relevant concentrations of compounds such as isoeugenol, making it a valuable tool for certain analytical assessments. Considering acceptable daily intake levels, isoeugenol in nutmeg does not pose a risk to human health. Similarly, the risk of isoeugenol in coffee must be estimated as low based on the available results.

Finally, initial studies on the simultaneous analysis of relevant phenylpropanoids by NMR were carried out. Signals of eugenol, methyleugenol, estragole, safrole as well as α- and β-asarone were identified and promising results were achieved. Once a full validation has been conducted, ^1^H NMR spectrometry is a practical and rapid method for the simultaneous analysis of phenylpropanoids in herbs and spices.

## Figures and Tables

**Figure 1 foods-13-00720-f001:**
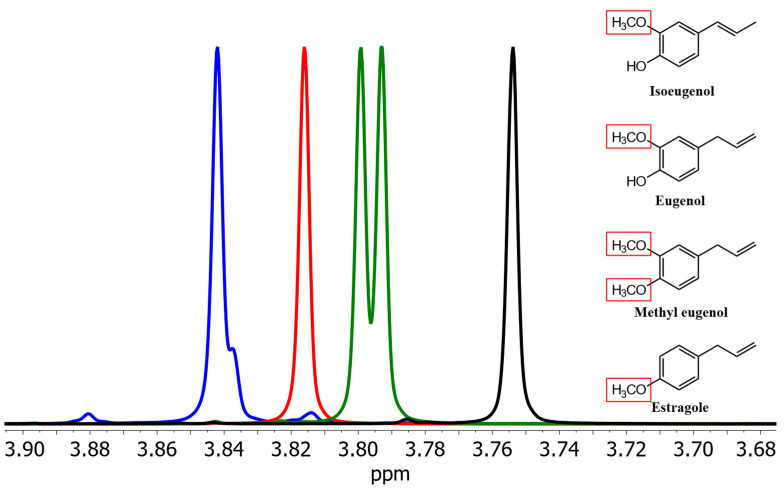
Section of the signals of the methoxy groups of isoeugenol (blue), eugenol (red), methyleugenol (green), and estragole (black) in (^2^H_4_)methanol. Red boxes highlight the methoxy group(s) causing the shown NMR resonances.

**Figure 2 foods-13-00720-f002:**
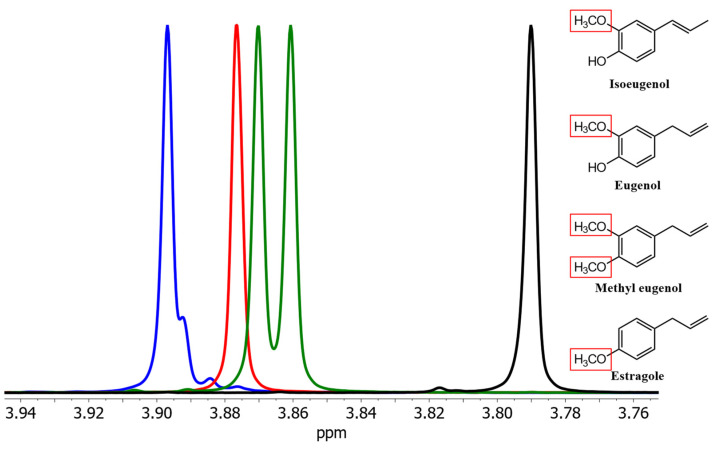
Section of the signals of the methoxy groups of isoeugenol (blue), eugenol (red), methyleugenol (green), and estragole (black) as pure spectra in (^2^H)chloroform. Red boxes highlight the methoxy group(s) causing the shown NMR resonances.

**Figure 3 foods-13-00720-f003:**
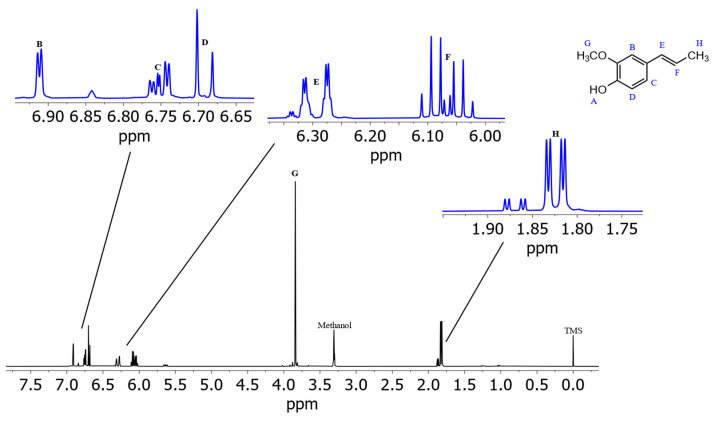
Spectrum of isoeugenol in (^2^H_4_)methanol with assigned signals of protons. Note: the proton of the hydroxy group (A) causes no resonance signal in (^2^H_4_)methanol due to hydrogen/deuterium exchange. Isoeugenol ^1^H NMR (D_3_COD, ISTD: TMS, 400 MHz, 300 K): *δ*_H_ 6.91 (d, 1H, *J* = 1.8 Hz), 6.75 (dd, 1H, *J* = 8.2 Hz, 1.8 Hz), 6.69 (d, 1H, *J* = 8.2 Hz), 6.29 (dq, 1H, *J* = 15.6 Hz, 1.6 Hz), 6.07 (dq, 1H, *J* = 15.6 Hz, 13.2 Hz, 6.6 Hz), 3.84 (3H, s), 1.86 (dd, 3H, *J* = 6.6 Hz, 1.6 Hz).

**Figure 4 foods-13-00720-f004:**
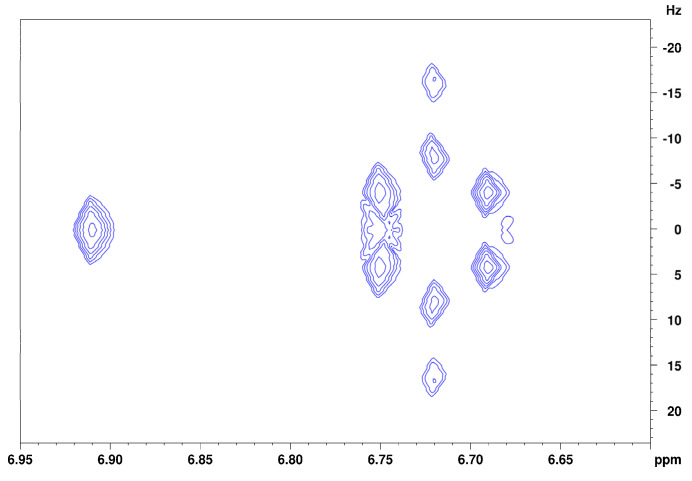
JRES spectrum of the aromatic region of isoeugenol in (^2^H_4_)methanol.

**Figure 5 foods-13-00720-f005:**
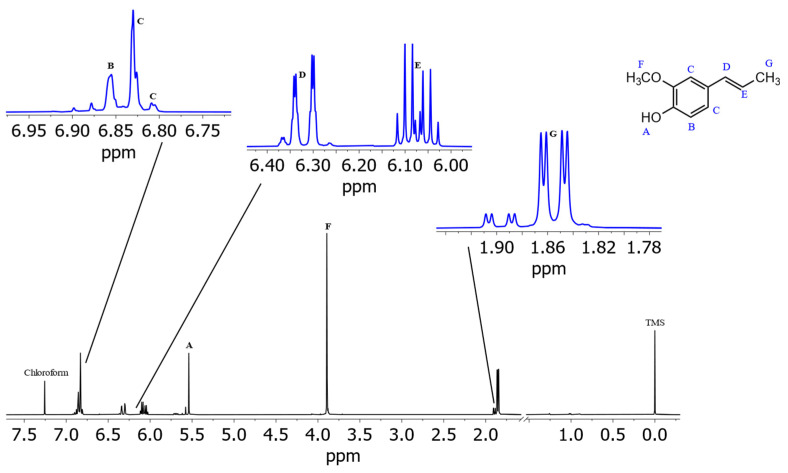
Spectrum of isoeugenol in (^2^H)chloroform with assigned signals of protons. Isoeugenol ^1^H NMR (CCl_3_D, ISTD: TMS, 400 MHz, 300 K): *δ*_H_ 6.86 (m, 1H), 6.83 (m, 2H), 6.32 (dq, 1H, *J* = 15.7 Hz, 1.6 Hz), 6.07 (dq, 1H, *J* = 15.7 Hz, 13.2 Hz, 6.6 Hz), 5.53 (s, 1H), 3.90 (3H, s), 1.86 (dd, 3H, *J* = 6.6 Hz, 1.6 Hz).

**Figure 6 foods-13-00720-f006:**
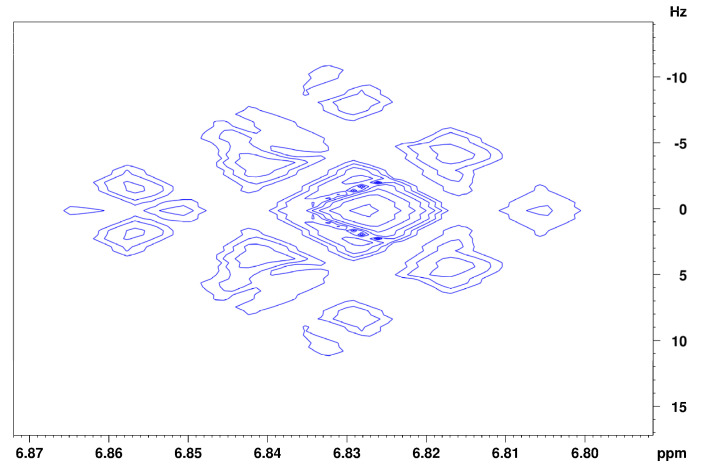
JRES spectrum of the aromatic region of isoeugenol in (^2^H)chloroform.

**Figure 7 foods-13-00720-f007:**
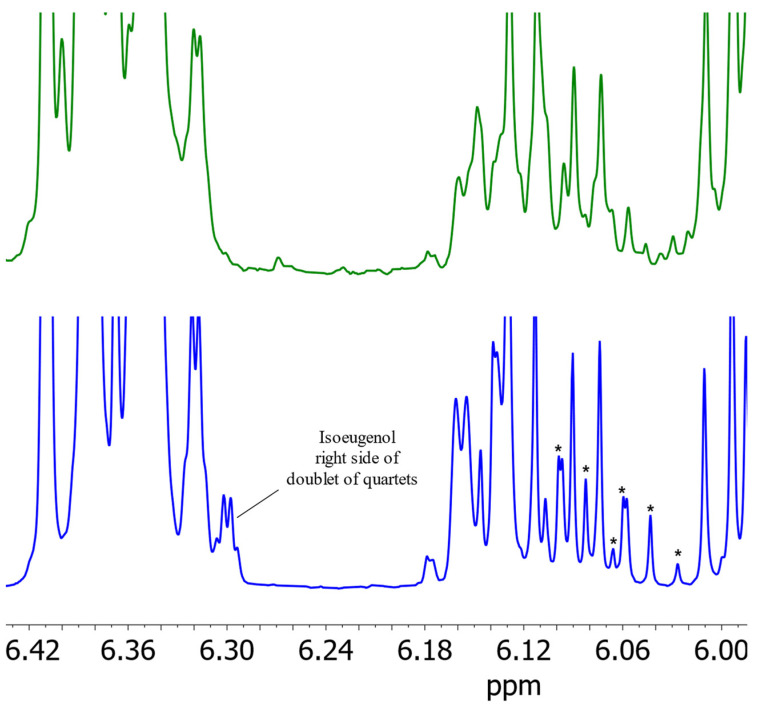
Comparison of sections of the ^1^H NMR spectra of nutmeg with the modified parameters of DIN EN ISO 6571 [18] for the production of essential oils by steam distillation (blue) and solvent extraction (green) in (^2^H)chloroform. The stars mark the identifiable signals of proton E (doublet of quartets).

**Figure 8 foods-13-00720-f008:**
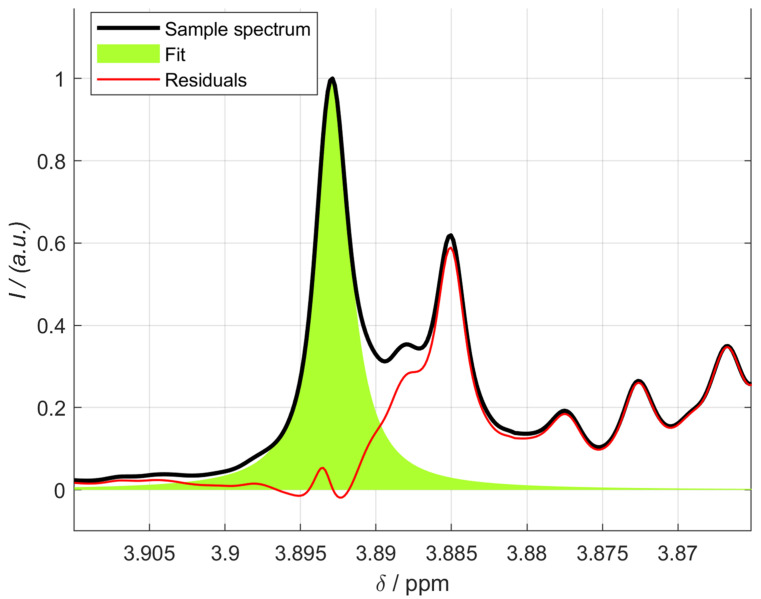
Integration of the isoeugenol singlet at 3.89 ppm of sample 1 using the MATLAB script.

**Table 1 foods-13-00720-t001:** Measurement parameters of the 400 MHz NMR spectrometer for the solvents (^2^H)chloroform and (^2^H_4_)methanol.

Parameter	(^2^H)Chloroform	(^2^H_4_)Methanol
pulse program	zg	noesygppr1d_d7
TD	65,788	65,536
SI	131,072	131,072
NS	32	32
DS	4	4
D1 [sec]	60	5
D7 [sec]	0	40
SW [ppm]	20.5617	20.5617
RG	22.6	22.6
O1P [ppm]	6.175	4.863
O1 [Hz]	2469.86	1945.15
AQ [sec]	3.9999104	3.9845889
TE [K]	300	300

TD (size of FID); SI (size of real spectrum); NS (number of scans); DS (number of dummy scans); D (delay); SW (spectral width); RG (receiver gain); O1P/O1 (transmitter frequency offset); AQ (acquisition time); TE (temperature).

**Table 2 foods-13-00720-t002:** Selected signals for the quantification of isoeugenol in (^2^H)chloroform and (^2^H_4_)methanol with the respective range of signals used in the MATLAB script.

Extracting Agent	Range of Signal [ppm]	Centre [ppm]	Multiplicity	Number of Protons
(^2^H)Chloroform	3.86–3.91	3.8965	s	3
(^2^H)Chloroform	6.27–6.33	6.2883	dd	0.5465
(^2^H_4_)Methanol	3.82–3.87	3.8410	s	3
(^2^H_4_)Methanol	6.24–6.31	6.2745	dd	0.5527

s (singlet); dd (doublet of doublet). Note: the number of protons of the signals from the doublet of doublets is given by the fact that only the right side of the signal was used for quantification since no overlaps occurred here for a large part of the sample matrices.

**Table 3 foods-13-00720-t003:** Summary of the limits of detection (LOD) and limits of quantification (LOQ) of the matrices coffee, calamus, nutmeg, and nutmeg essential oil according to DIN 32645:2008 [22].

Matrix	Extraction Agent	LOD [mg/kg]	LOQ [mg/kg]
Coffee	(^2^H)Chloroform	20	47
Calamus	(^2^H)Chloroform	160	386
Nutmeg	(^2^H_4_)Methanol	191	468
Nutmeg essential oil	(^2^H)Chloroform	139	340

**Table 4 foods-13-00720-t004:** Contents of isoeugenol in the purchased as well as prepared essential oils of nutmeg with the respective signal used for quantification.

Essential Oil Nutmeg	Centre of Signal	Isoeugenol [g/kg]
Sample 1	s (3.89 ppm)	11.2 ± 0.10
Sample 2	dd (6.30 ppm)	3.68 ± 0.09
Sample 3 *	dd (6.30 ppm)	8.36 ± 0.06
Sample 4 *	s (3.89 ppm)	8.42 ± 0.14

s (singlet); dd (doublet of doublet); n = 3. Note: the samples marked with a star were obtained via steam distillation from plant material.

## Data Availability

The data presented in this study are available on request from the corresponding author. The data are not publicly available due to institutional policy regarding data protection.

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
