# Peer review of "Quantitative NMR Spectrometry of Phenylpropanoids, including Isoeugenol in Herbs, Spices, and Essential Oils"

_foods, 2024, doi:10.3390/foods13050720_

Round 1

Reviewer 1 Report

Comments and Suggestions for Authors

The research is of great interest in the aim of valorize NMR spectroscopy as a potential tool for food safety evaluation, considering that this method can present some advantages (but also disadvantages) respect of other analysis methods.

Anyway, there are some concerns that have to be better discussed.

First of all, it is important in introduction section, to introduce a general comparison between NMR and chromatographic analyses in the field of detect and quantify specific compounds for analytical studies. It is noteworthy that chromatographic analyses are characterized by a higher sensitivity respect to NMR but, on the other side, NMR cis characterized by advantages namely, reproducibility, time of analysis, etc. Authors should make a comparison between the two approaches and explain why the use of NMR can be useful, in some cases, in analytic studies for food quality and safety.

The experimental design is very clear and well described in all the aspects. Just an observation: considering that 1300 samples of coffee have been analyzed, their preparation for analysis should be mentioned in the text.

Results and discussion: this section is very long, mainly considering that isoeugenol is measured only in essential oil. First of all, I suggest resuming the obtained data for each matrix in a table, in order to give an easy and simplified way to read the results.

Moreover, although the main focus is the isoeugenol detection and quantification, most of the analyzed matrices are discussed on the base of other detected molecule. For this reason, the title of the paper should be changed with a more general aim of the research (for example, Phenylpropanoids in herbs, species and essential oils). The other possibility is to strongly reduce the discussion section regarding the chemical profile of matrices where isoeugenol was not detected. 

422: although the explanation is present in Chapter 3.1.5. as indicated, a brief explanation should be done to make the reading more fluid.

Title of “3.7 Toxicological Evaluation of Isoeugenol” section should be changed since it is more suitable for effective toxicological studies (biological). I think that these data can be omitted of just moved in the discussion section regarding each corresponding matrix.

At the end of the discussion section, a discussion regarding the potentiality of the NMR methos should be carried out, underlying both positive and negative aspects of the approach.

The presence of references in conclusion section is not a good practice since this paragraph should be dedicated to final considerations and perspectives. I suggest to eliminate references and further emphasize the usefulness of the developed NMR approach and the potential perspectives.

Author Response

The research is of great interest in the aim of valorize NMR spectroscopy as a potential tool for food safety evaluation, considering that this method can present some advantages (but also disadvantages) respect of other analysis methods.

Anyway, there are some concerns that have to be better discussed.

RESPONSE: Thank you for the evaluation of our paper.

First of all, it is important in introduction section, to introduce a general comparison between NMR and chromatographic analyses in the field of detect and quantify specific compounds for analytical studies. It is noteworthy that chromatographic analyses are characterized by a higher sensitivity respect to NMR but, on the other side, NMR cis characterized by advantages namely, reproducibility, time of analysis, etc. Authors should make a comparison between the two approaches and explain why the use of NMR can be useful, in some cases, in analytic studies for food quality and safety.

RESPONSE: A sentence was added around line 67 specifying the differences between the techniques.

The experimental design is very clear and well described in all the aspects. Just an observation: considering that 1300 samples of coffee have been analyzed, their preparation for analysis should be mentioned in the text.

RESPONSE: This was a simple chloroform extraction. The info was added to the text, but the full method is cited in the literature. The procedure also conforms to the draft standard DIN EN 17992.

Results and discussion: this section is very long, mainly considering that isoeugenol is measured only in essential oil. First of all, I suggest resuming the obtained data for each matrix in a table, in order to give an easy and simplified way to read the results.

RESPONSE: We are unsure what might be placed into a table, as many matrices were negative.

Moreover, although the main focus is the isoeugenol detection and quantification, most of the analyzed matrices are discussed on the base of other detected molecule. For this reason, the title of the paper should be changed with a more general aim of the research (for example, Phenylpropanoids in herbs, species and essential oils). The other possibility is to strongly reduce the discussion section regarding the chemical profile of matrices where isoeugenol was not detected.

RESPONSE: Phenylpropanoids was added to the title.

422: although the explanation is present in Chapter 3.1.5. as indicated, a brief explanation should be done to make the reading more fluid.

RESPONSE: We decided to delete the cross-reference to have the discussion in only one instance.

Title of “3.7 Toxicological Evaluation of Isoeugenol” section should be changed since it is more suitable for effective toxicological studies (biological). I think that these data can be omitted of just moved in the discussion section regarding each corresponding matrix.

RESPONSE: The title of the section was changed to avoid confusion with in vivo or in vitro toxicology studies. Otherwise, we do not want to scatter this discussion into other sections of the paper.

At the end of the discussion section, a discussion regarding the potentiality of the NMR methos should be carried out, underlying both positive and negative aspects of the approach.

RESPONSE: We have included this point in the revised version of the conclusion section. Otherwise, we wanted to avoid textbook knowledge in the discussion section, as the topic of our paper was not a methodological comparison of NMR and chromatography.

The presence of references in conclusion section is not a good practice since this paragraph should be dedicated to final considerations and perspectives. I suggest to eliminate references and further emphasize the usefulness of the developed NMR approach and the potential perspectives.

REPONSE: Thank you for the suggestion. We have done this and made the conclusion more concise, and less repetitious in general.

Reviewer 2 Report

Comments and Suggestions for Authors

With interest I have read your manuscript. It is in general ready to be published once the following issues have been clarified:

Two minor concerning typography and grammar:

Line 55: replace the semicolons by commas.

L109: Replace "For" by To.

The explanation of the treatment for nutmeg is confusing. On L316ff it appears as if all nutmeg must be steam disitilled for proper analysis. Then on L374ff you describe solvent extraction for 11 samples and on L416ff the analysis of the essential oils of 12 samples of which, apparently, you distilled three yourself. It is not clear why you chose to ony distil three even though you point to the fact that solvent extraction  is suboptimal. Please clarify.

L360ff: All materials in this section are solids except nutmeg essential oil. Therefore, stating recoveries in concentration units is not meaningful and does not allow relation to the LOD/LOQ values of Table 3. In addition, the SI unit for volume is a upper-case L, not lower-case. Revise.

Author Response

With interest I have read your manuscript. It is in general ready to be published once the following issues have been clarified:

RESPONSE: Thank you for the evaluation of our paper.

Two minor concerning typography and grammar:

Line 55: replace the semicolons by commas.

RESPONSE: The semicolons were replaced as requested.

L109: Replace "For" by To.

RESPONSE: Thanks. The replacement was done.

The explanation of the treatment for nutmeg is confusing. On L316ff it appears as if all nutmeg must be steam disitilled for proper analysis. Then on L374ff you describe solvent extraction for 11 samples and on L416ff the analysis of the essential oils of 12 samples of which, apparently, you distilled three yourself. It is not clear why you chose to ony distil three even though you point to the fact that solvent extraction  is suboptimal. Please clarify.

RESPONSE: There is no discrepancy. The statements are correct. In L374 only the other components but not isoeugenol was detected. In L416 it is correct that the distilled only 3 samples, because the amount of material was not sufficient to self-distill all samples. We have clarified the sample numbers in the respective section of materials and methods.

L360ff: All materials in this section are solids except nutmeg essential oil. Therefore, stating recoveries in concentration units is not meaningful and does not allow relation to the LOD/LOQ values of Table 3.

RESPONSE: All values are now stated recalculated to mg/kg to make the data presentation consistent (note: liquid samples were also weighed, so that data per kg and not only per L can be reported).

In addition, the SI unit for volume is a upper-case L, not lower-case. Revise.

RESPONSE: The unit was now deleted and re-calculated to mg/kg (see previous response).

Reviewer 3 Report

Comments and Suggestions for Authors

In this manuscript, the authors studied the presence of isoeugenol in herbs, spices, and essential oils employing NMR spectrometry. The study is interesting, the methodology is adequate, and the discussion is consistent with the results presented. The manuscript has merits, so in my view, the manuscript can be published in Foods journal after minor reviews as described below.

The introduction is well-referenced. However, the objective of the work is not clear, I suggest the authors reinforce the reasons for developing 1H NMR quantitative analysis if simpler methods such as GC-MS is commonly used to qualify and quantify eugenol.

The methodology is feasible and well-referenced. Some points must increase in quality in describing the procedure.

On p.2, please insert the exact origin (commercial brand, tracking number, …) or exsiccate register of each sample.

On p. 3 please insert the brand and origin of the deuterated solvents used.

Results and discussion are concise.

Reference patterns must be standardized, please revise all reference section. 

Comments on the Quality of English Language

Minor editing of English language required.

Author Response

In this manuscript, the authors studied the presence of isoeugenol in herbs, spices, and essential oils employing NMR spectrometry. The study is interesting, the methodology is adequate, and the discussion is consistent with the results presented. The manuscript has merits, so in my view, the manuscript can be published in Foods journal after minor reviews as described below.

The introduction is well-referenced. However, the objective of the work is not clear, I suggest the authors reinforce the reasons for developing 1H NMR quantitative analysis if simpler methods such as GC-MS is commonly used to qualify and quantify eugenol.

RESPONSE: The advantages of NMR were stressed in the introduction, see also response to reviewer #1 above.

The methodology is feasible and well-referenced. Some points must increase in quality in describing the procedure.

On p.2, please insert the exact origin (commercial brand, tracking number, …) or exsiccate register of each sample.

RESPONSE: All samples used in this study were commercially purchased and are not accompanied by specific authentication measures such as commercial brand identification, tracking numbers, or exsiccate registers. This approach reflects the study's focus on assessing the general presence and quantification of isoeugenol in commercially available products, rather than attributing findings to specific brands or batches. To clarify this, “commercially” was added in the sampling section of the manuscript.

On p. 3 please insert the brand and origin of the deuterated solvents used.

RESPONSE: The requested information was added.

Results and discussion are concise.

RESPONSE: Thank you for the assessment of our paper!

Reference patterns must be standardized, please revise all reference section. 

RESPONSE: The references were carefully revised to correspond to the authors’ guidelines.

Round 2

Reviewer 1 Report

Comments and Suggestions for Authors

Authors responded to all the issues and improved the quality of the paper. No criticism were present so the paper can be accepted in the present form

Author Response

Thank you for your assessment of our paper.

The following changes were made in response to  the editors remarks:

Reviewers comments improved significantly the manuscript quality and authors responded to all the issues which is greatly appreciated. However still two minor aspects need to be improved before been accepted for publication, namely

  1. i) to review text to replace numbers 1 to 9 by words (example replace 1 sample by one sample, line 86, 2 samples by two samples, etc.) and

RESPONSE: All numbers were replaced as requested.

  1. ii) references still need to be improved since several aspects are not homogeneous as:
    ref 40-2x2014

RESPONSE: This is actually correct. The journal is using the year as volume number. Therefore the year field is 2014 and the volume number field is 2014.

ref 36, 37,32 and others- please homogenize the signal after the data which is different: sometimes it is ., other , other;

RESPONSE: The fields were homogenized. We actually used the MDPI style of Citavi and currently cannot explain the problems. Hopefully, the copy-editor can deal with potentially remaining issues.

the same to doi, authors need to homogenize if use . or , before doi
RESPONSE: The use of doi was corrected throughout.

ref 67 at least but check for others: review and homogenize the place of date in the reference

RESPONSE: Done.